# Potential Applications of *Arthrospira platensis* Lipid-Free Biomass in Bioremediation of Organic Dye from Industrial Textile Effluents and Its Influence on Marine Rotifer (*Brachionus plicatilis*)

**DOI:** 10.3390/ma14164446

**Published:** 2021-08-08

**Authors:** Ahmed E. Alprol, Ahmed M. M. Heneash, Mohamed Ashour, Khamael M. Abualnaja, Dalal Alhashmialameer, Abdallah Tageldein Mansour, Zaki Z. Sharawy, Mouhamed A. Abu-Saied, Abd El-Fatah Abomohra

**Affiliations:** 1National Institute of Oceanography and Fisheries (NIOF), Cairo 11516, Egypt; ah831992@gmail.com (A.E.A.); aheneash@yahoo.com (A.M.M.H.); zaki_sharawy@yahoo.com (Z.Z.S.); 2Department of Chemistry, College of Science, Taif University, P.O. Box 11099, Taif 21944, Saudi Arabia; k.ala@tu.edu.sa (K.M.A.); Dsamer@tu.edu.sa (D.A.); 3Animal and Fish Production Department, College of Agricultural and Food Sciences, King Faisal University, P.O. Box 420, Al-Ahsa 31982, Saudi Arabia; 4Fish and Animal Production Department, Faculty of Agriculture (Saba Basha), Alexandria University, Alexandria 21531, Egypt; 5Polymeric Materials Research Department, Advanced Technology and New Materials Research Institute, City of Scientific Research and Technological Applications (SRTA-CITY), New Borg El-Arab City, Alexandria 21934, Egypt; mouhamedabdelrehem@yahoo.com; 6New Energy and Environmental Laboratory (NEEL), School of Architecture and Civil Engineering, Chengdu University, Chengdu 610106, China; 7Botany and Microbiology Department, Faculty of Science, Tanta University, Tanta 31527, Egypt

**Keywords:** cyanobacterium, *Arthrospira platensis* NIOF17/003, bioremediation, Ismate violet 2R, adsorbent isotherm models, FTIR, rotifer bioassay

## Abstract

*Arthrospira platensis* is one of the most important cultured microalgal species in the world. *Arthrospira* complete dry biomass (ACDB) has been reported as an interesting feedstock for many industries, including biodiesel production. The *A. platensis* by-product of biodiesel production (lipid-free biomass; LFB) is a source of proteins, functional molecules, and carbohydrates, and can also be reused in several applications. The current study investigated the efficiency of ACDB and LFB in bioremediation of dye (Ismate violet 2R, IV2R) from textile effluents. In addition, the potential of ACDB and LFB loaded by IV2R as a feed for Rotifer, *Brachionus plicatilis*, was examined. The surface of the adsorbents was characterized by SEM, FTIR, and Raman analysis to understand the adsorption mechanism. The batch sorption method was examined as a function of adsorbent dose (0.02–0.01 g L^−1^), solution initial concentration (10–100 mg L^−1^), pH (2–10), and contact time (15–180 min). The kinetic studies and adsorption isotherm models (Freundlich, Langmuir, Tempkin, and Halsey) were used to describe the interaction between dye and adsorbents. The results concluded that the adsorption process increased with increasing ACDB and LFB dose, contact time (120 min), initial IV2R concentration (10 mg L^−1^), and acidity pH (2 and 6, respectively). For the elimination of industrial textile wastewater, the ACDB and LFB sorbents have good elimination ability of a dye solution by 75.7% and 61.11%, respectively. The kinetic interaction between dye and adsorbents fitted well to Langmuir, Freundlish, and Halsey models for LFB, and Langmuir for ACDB at optimum conditions with R^2^ > 0.9. In addition, based on the bioassay study, the ACDB and LFB loaded by IV2R up to 0.02 g L^−1^ may be used as feed for the marine Rotifer *B. plicatilis*.

## 1. Introduction

The release of untreated sewage waste and industrial emissions endangers marine ecosystems [1]. The effluents of textile industries are a complex, heavily colored, photolytically stable, and highly instable mixture of several contaminating materials; these substance are rich in dyes and other molecules, which induce color combined with the organic load conducing the breakdown of the whole ecological balance of the receiving aquatic system because they are very stable in natural environments [2]. The majorities of dyes are a possible health danger to all forms of life with long-term and occasional overexposure such as eczema, skin dermatoses, and allergic responses, and may affect the lungs, liver, immune system, vasco-circulatory system, and reproductive systems for animals and humans. In addition, they have a great effect on the photosynthetic activity of marine biota [3]. According to Mondal et al. [4], the standard and allowable levels of various parameters in textile industry wastewater must be 5.5–9 for pH, 30 mg L^−1^ and 250 mg L^−1^ for BOD and COD parameters, respectively, in addition to 100 mg L^−1^ and 500 mg L^−1^ for TSS and TDS, while 1 mg L^−1^ for total residual chlorine is included. Permissible levels will vary by regulatory agency and municipality, according to information published by the American Water Works Association and the Water Pollution Control Federation of the United States [5], which indicated that any fungicide, pesticide, insecticide, rodenticide, herbicide, or fumigant discharged into any watercourse must not include any of the following issues as radioactive material: sawdust, garbage, timber, refuse, human or animal waste or solid matter, petroleum and/or other flammable solvent. Furthermore, the allowed effluent discharge limitations in controlled watercourses must not exceed around 7 Lovibond units for color, 5 mg L^−1^ for detergents (linear alkylate sulphonate such as methylene blue active chemicals), 1 mg L^−1^ (total) for grease and oil, and 0.5 mg L^−1^ for metals in total [5]. Turbinaria conoides was used by Rajeshkannan et al. [6] to study the biosorption of malachite green. Malachite green biosorption was detected to be highest at 30 °C and pH 8.0.

Textile dye effluent treatment techniques are becoming more difficult owing to increasing the industrial effluents disposal costs, diminishing resources of water, and stricter discharge rules that need decreased acceptable pollutant levels in wastewater streams; therefore, it is important to remove these dyes from industrial effluents before discharging into the environment [7]. Several physiochemical treatment methods for dye elimination have been investigated, including ozonation, the adsorption membrane process, electrodialysis, coagulation–flocculation, nano-filtration, trickling filters, membrane processes, reverse osmosis, and adsorption techniques [8]. Adsorption is one of the effective equilibrium separation technologies that so far have been successful in removing pollutants from wastewater owing to its simple design, high adsorption capacity, simplicity of operation, insensitivity, and flexibility to toxic pollutants [9].

Algal biomass is an attractive inbred bioactive substances that can be used in different industrial uses including aquaculture diet feed additive [10], aquaculture water-conditioner [11], plant growth biostimulants [12], the food industry [13], pharmaceuticals [14], antimicrobial substances [15], cosmetics [16], bioremediation [17], biodiesel [18,19], and crude bio-oil [20]. One of the most interesting applications is the production of biodiesel from microalgae due to the relatively high lipid productivity [18,19]. For many considerations, algal biomass, in the near future, will be the most important sustainable feedstock for the production of biodiesel, attributed to its high lipid percentage, which can be increased more than 40-fold greater than those of other plants [21]. Interestingly, the residual of the algal cells after oil extraction, i.e., lipid-free biomass, contains many bioactive compounds that can be used in different applications, such as aquafeeds, animal diet, or bioactive materials for pharmaceutical and cosmetic products [18].

The cyanobacterium filamentous blue green microalga *Arthrospira platensis* NIOF17/003 has various promising biotechnological applications. Over the world, this species is one of the most microalgal species that have great attention and have been widely cultured on a commercial scale [14,22]. The commercial production of *A. platensis* is generally developed around the world and increased from 48,000 tons in 2005 to over 89,000 tons in 2016 [23]. To develop strong commercial applications of native aquatic microorganisms, a scientific database of isolation, screening, molecular identification, growth conditions, biochemical composition, and bioactive substances must be created [19,22]. On the other hand, zooplanktons are considered as an important indicator for determining water pollution [24]. Moreover, many studies reported that the microalgal lipid-free biomass as a by-product from biodiesel production is a good source as feed for zooplanktons such as Artemia [19,25] and rotifer [22]. In addition, *A. platensis* contains many bioactive compounds which have different functional groups such as phosphate, hydroxyl, sulfate, carboxyl, and other charged groups which can intervene in pollutant binding. Therefore, it has been effectively utilized to eliminate toxic heavy metals and dyes from aqueous solutions [26].

Ismate violet 2R was chosen as a model compound in this research because of its wide variety of applications, which include dyeing silk, cotton, rayon, leather, paper, wood, and coating for paper stock and medical purposes, as well as its potential harm [27]. Furthermore, the Ismate violet 2R was classified as a sulfur dye based on their application, which is highly soluble in aqueous solution and has a higher negative charge density, as well as their adsorption preference for various adsorbents [28]. This could indicate that the adsorption process was influenced by the electrical connection between positively charged adsorbent surfaces and anionic dyes. Aromatic materials can be evaluated with sodium sulfide and re-oxidized to insoluble sulfur-containing chemicals on fibers as a sulfur coloring application method [29]. However, the application of lipid-free *A. platensis* biomass for dye removal and its consequent impact on rotifers were not previously evaluated. Consequently, the present study was conducted to investigate the phytoremediation possible of *Arthrospira* complete dry biomass (ACDB) and lipid-free biomass (LFB) of *Arthrospira platensis* NIOF17/003 (GenBank accession number: MW396472) in the removal capacity of Ismate violet 2R dye (IV2R) from textile effluents, either at a laboratory or pilot scale. Alternative kinetic adsorption and equilibrium isotherm models were used to determine the removal capacities of adsorbents (ACDB and LFB). Additionally, the current study investigated the bioassay and potential application of the adsorbents (ACDB and LFB) enriched with IV2R as a primary feedstock for the marine Rotifer, *B. plicatilis*.

## 2. Materials and Methods

### 2.1. Arthrospira Platensis NIOF17/003

#### 2.1.1. Algal Strain and Oil Extraction

*Arthrospira platensis* NIOF17/003 adsorbent, isolated from El-Khadra alkaline-salty Lake in Wadi El-Natrun city, Egypt, (30°13′546″ E; 30°26′504″ N), has been recorded at the international GenBank with accession number: MW396472 [22]. *A. platensis* NIOF17/003 indoor cultivation, growth curve determination, harvesting, biochemical composition, lipid extraction, and biodiesel production were achieved as defined by Zaki et al. [22]. *A. platensis* NIOF17/003 was batch cultured for twelve days in 500 mL sterilized Zarrouk medium under controlled growth conditions of continuous illumination (3500–4000 Lux), temp (28.5 ± 2 °C), and incessant sterile aeration, with shaking at 80 rpm. At late exponential phase (day 8), the biomass was obtained by centrifugation (7000× *g* for 10 min^−1^). Biochemical composition, lipid extraction, lipid productivity, and the physicochemical properties of FAMEs of the biodiesel of *A. platensis* NIOF17/003 were determined and calculated as described by Zaki et al. [22]. After oil extraction, *Arthrospira* complete dry biomass (ACDB) and lipid-free biomass (LFB) of *A. platensis* NIOF17/003 were dried for 48 h at 55 °C and conserved at 10 °C for additional characterizations.

#### 2.1.2. Characterization of ACDB and LFB

The dried *A. platensis* NIOF17/003 biomass was milled by (Wiley Mill Standard, 03, USA) then sieved to minor particle size with the average of 75 µm, then suspended in distilled water and vortexed for 20 min at 10,000 rpm (Dremel, 1100-01, Brazil). These conditions were created by the previous preliminary examinations; in addition, the size distribution was not affected by the pH. The morphological investigations of the adsorbents (ACDB and LFB) were carried out by scanning electron microscope (SEM), (JEOL JSM 6360 LA). Furthermore, Fourier transform infrared (Shimadzu FTIR-8400 S, Kyoto, Japan) and Raman analysis spectrophotometry (a Senterra Raman spectrometer, USA) were used to measure the influence of the dye on biomass.

### 2.2. Dye Solution Preparation

Stock solution of the dye Ismate violet 2R (IV2R) was archived by dissolving the accurately balanced dye in distilled water at 1000 mg L^−1^ of concentration, without additional purification. The specification of IV2R is presented in Table 1. The investigational mixtures were achieved by diluting the IV2R dye stock mixture to the accurate proportions to obtain various preliminary concentrations.

### 2.3. Adsorption Experiments

The effects of contact time (15, 30, 45, 60, 120, and 180 min), pH (2, 4, 6, 8, and 10), initial IV2R dye concentration (10, 20, 30, 40, 50, and 100 mg L^−1^) [31], and different adsorbent concentrations (0.02, 0.04, 0.06, 0.08, and 0.1 g L^−1^) of either ACDB or LFB were examined for the IV2R removal. The particular conditions have been cited in the related plots. The experiments were performed in triplicates by batch technique to find the equilibrium data. The reaction mixture containing 10 mL of dye solution was shacked at room temperature (25 ± 2 °C) at 110 rpm under the specified conditions. For these experiments, amounts of dye solutions (10 mg L^−1^) were fixed and selected in the study of the effects of pH, contact time, and adsorbent doses. The concentrations of dye in the solution were assessed quantitatively, as stated by Lambert–Beer law [28], by linear equations achieved through plotting a calibration curve for IV2R dye at concentrations series and absorbance at 550 nm. Both of adsorption capacity (q_e_) and dye removal percentages were calculated as described previously [26] by the following equations:(1)$$\mathrm{Percentage}\;\mathrm{removal}\;\left(\%\right)=\frac{\left(C_{i}-C_{f}\right)}{C_{i}}\times 100$$
(2)$$\mathrm{Adsorption}\;\mathrm{capacity}\;(q_{e})=\frac{\left(C_{i}-C_{f}\right)\times \;V}{W}$$
which: C_i_ and C_f_ are the primary concentration at initial time and the final concentration IV2R at certain period (mg L^−1^), respectively, while V is the volume of the dye mixture (L); in addition, W is the weight of the dry adsorbent (g).

### 2.4. Adsorption Isotherm Experiments

The extent of elimination of IV2R from aqueous solution strongly depends on the preliminary dye concentrations; therefore, different IV2R concentrations of 10–100 mg L^−1^ were investigated at constant parameters of pH 6, 30 °C, 3 h, with 0.2 g of ACDB and LFB, at 150 rpm which was adequate to reach equilibrium [26]. The data was fitted and calculated into the following isotherm experiments: Langmuir, Freundlich, Tempkin, and the Halsey model.

#### 2.4.1. Langmuir Model

The Langmuir model is presented by the following a mathematical expression [32]:q_e_ = q_max_ bCe/(1 + bCe)(3)
where: q_max_ is the higher sorption capacity (mg g^−1^) consistent to the saturation capacity (representing whole binding sites of adsorbent) and b is the coefficient regarding to the affinity among biomass and dyes ions (L mg^−1^); in addition, b is the energy of the sorption process.

The relationship of this model can be achieved from plotting curve (1/q_e_) vs. (1/C_e_):1/qe = 1/(bq_max_ C_e_) + 1/q_max_(4)
where b and q_max_ are calculated from slope and intercept of the linear equation, respectively.

#### 2.4.2. Freundlich Model

For the examinable the ability of Freundlich model to fit the experimental data, via plotting a curve of log q_e_ with respect to log C_e_ was used to create the slope of n and the intercept value of K_f_. Moreover, the Freundlich model could be definitely linearized through plotting it in the following logarithmic Equation [33]:log q_e_ = logK_f_ + 1/n logC(5)

Freundlich constants K_f_ and n have been determined by the isotherm equation according to Equation (5).

#### 2.4.3. Temkin Model

Tempkin linear isotherm is given as follows [34]:q_e_ = B ln A + B ln C_e_(6)
where: B = RT/b, A shows the equilibrium binding constant (Lg^−1^) correlated to the higher binding energy, b (J mol^−1^) is a constant corresponding to heat of adsorption, while B = (RT/b) (J mol^−1^) is the Tempkin constant and the heat of the sorption process, R is the gas constant (8.314 J mol^−1^ K), and T is the absolute temperature (Kelvin). The sorption results can be analyzed according to Tempkin Equation (6). Therefore, a plot curve of q_e_ vs. ln C_e_ allows the examination of the Tempkin isotherm constants A and B from the slope and intercept.

#### 2.4.4. The Halsey Isotherm Model

The Halsey equation is appropriate for a multilayer sorption process; in addition, the fitting of the Halsey model can be heteroporous solids [35]. The Halsey model could be applied in the following Equation:(7)$$\mathrm{Ln}\;q_{e}=\frac{1}{n}\mathrm{Ln}\;K+\frac{1}{n}\;\mathrm{Ln}\;C_{e}$$
where: K and n are Halsey constants.

### 2.5. Adsorption Kinetics

Kinetic studies are achieved in 50 mL conical flasks at liquid of pH 2, 0.1 g of adsorbent was mixed separately with 50 mL of IV2R mixture of 10 mg L^−1^ concentrations and the solution was agitated at room temperature under requisite time intervals viz. 15, 30, 45, 60, 120, and 180 min.

#### 2.5.1. Pseudo-First Order Kinetic

The linear form of the generalized of pseudo-first order equation [34] is given by the following equation:dq/d_t_ = K_1_ (q_e_ − q_t_)(8)
where: q_e_ (mg g^−1^) is the amount of dye adsorbed at equilibrium, q_t_ (mg g^−1^) is the quantity of dyes adsorbed at time t, and K_1_ (min^−1^) is expressed as the pseudo first-order rate constant. The integrating equation is given by the following equation:log (q_e_/q_e_ − q_t_) = k_1_ t/2.303(9)

The equation of pseudo-first order is given by the following formula in the linear equation:log (q_e_ − q_t_) = log q_e_ − K_1_t/2.303(10)

The q_e_ and K_1_ values were calculated from the intercept and slope of the linear plots of log (q_e_ − q_t_) verses t at an initial dye concentration of 10 mg L^−1^.

#### 2.5.2. Pseudo-Second Order Kinetic Model

The pseudo-second order equation was expressed as following [36]:dq_t_/d_t_ = k_2_ (q_e_− q_t_)^2^(11)
where: k_2_ (g mg^−1^ min^−1^) indicates the pseudo-second order rate constant. The integrating equation for the boundary conditions of t = 0 to t and, correspondingly, q_t_ = 0 to qt presented:1/(q_e_ − q_t_) = 1/q_e_ + k_2_(12)

The equation of the pseudo-second order was adjusted to obtain a linear form as described by Ozacar and Şengil [37] as the following:t/q_t_ = 1/k_2_ q_e_^2^ + t/q_e_(13)

Plots of (t/q_t_) against (t) can provide a linear correlation from which the values of parameters q_e_ and K_2_ can be calculated from the slope and intercept, respectively.

#### 2.5.3. The intraparticle Diffusion Model

The intraparticle diffusion equation is explored by the following equation:q_t_ = K_dif_ t^1/2^ + C(14)
where: C is the intercept; in addition, K_dif_ (mg g^−1^ min^−^^0.5^) reflects the intraparticle diffusion rate constant.

### 2.6. Application of ACDB and LFB on Actual Wastewater

In order to detect the efficiency of ACDB and LFB to decolorize textile effluents, textile wastewater was collected in sterile containers from Misr company for textile dyeing and printing located in the Al-Mahala Al-Kobra (Gharbia, Egypt) and mixed with wastewater sample collected from El-Emoum drain, neighboring Lake Maruit, Alexandria, Egypt, which contains industrial sewage and agriculture waste. Deionized water including the analogous concentration of dyes was prepared to apply as a control to evaluate the influence of adsorbent on IV2R dye elimination. The application of ACDB and LFB to remove IV2R from wastewater was carried out using the optimum experimental conditions for each ACDB and/or LFB.

### 2.7. Bioassay Test

The harvested biomass (ACDB and/or LFB) that was loaded with IV2R was used as aqua-feed to determine its toxic potential on the marine Rotifera, *Brachionus plicatilis,* (L-type, mean length of 180 µm). Prior to the bioassay test, *B. plicatilis* were maintained and cultured under controlled conditions (23 °C, 30 ppt, pH 7.5, continuous thin aeration, while being supplemented with the native green microalga *Nannochloropsis oceanica* NIOF15/001 [19], at a density of 5.5 × 10^6^ cells ml^−1^ day^−1^. Rotifer individuals were collected from the culture tanks, starved for 24 h to reach complete gut discharge, and then distributed into plastic jars filled with 500 mL of filtered seawater, three replicates for each level. The bioassay experiment was conducted for 72 h under constant conditions (23 °C, 30 ppt, pH value 7.5, and without aeration). In the present study, the effect of different doses (0.02, 0.05, 0.10, and 0.20 g) of ACDB and LFB, those loaded with IV2R compared to the same levels without loading IV2R on rotifer population growth, rotifer mortality, female carry eggs population, and the mortality of female carry eggs were investigated. Using an optical microscope, the rotifer tested parameters were measured using a Sedgwick-Rafter counting cell as previously described [22]. The rotifer population was calculated as an increase or decrease of the individual total number while the initial stocking density of rotifer was 17,664 ± 1154 individual L^−1^. The rotifer carry eggs population was calculated as the number of rotifer female carry eggs, while the initial stocking density of rotifer female carry eggs was 14,500 ± 1850 females L^−1^. Moreover, total mortality and that of female carry eggs were investigated as dead individuals and females, respectively, during samples investigation under the microscope.

### 2.8. Statistical Analysis

The homoscedasticity suspicions, normality, and endogeneity of the presented data (mean ± standard deviation, SD, n = 3) were affirmed before the statistical analysis which was performed via the SPSS program (IBM, v. 20, Armonk, NY, USA). All estimated variables were applied, at a significant level (*p* < 0.05), to a study of variance (ANOVA), Duncan’s multiple range examinations, and the least significant difference (LSD) tests.

## 3. Results and Discussion

### 3.1. Adsorbents Characterization

#### 3.1.1. FTIR Analysis

Figure 1 shows the FTIR spectra of *A. platensis* (ACDB and LFB) before and after adsorption, which revealed broad, intense absorption peaks at 3271.8–3856.8 cm^−1^. It displays the stretching vibration of NH and O-H groups. In addition, the intense band observed at 2917 and 2922 cm^−1^ indicates the extending vibrations of asymmetric CH bond of methoxy, methylene, and methyl groups. The presence of peaks around 2349.79 cm^−1^ might be owing to presence of amide and thiol S-H stretching. Furthermore, the absorption peaks at 2067.80 cm^−1^ refer to (C≡C). The absorbance band at 1635.36 and 1628.89 cm^−1^ can be ascribed to the existence of carbonyl C=O stretching vibration of the carboxyl groups and aromatic C=C ring stretching. The intense band seen at 1540.36 and 1521.77 cm^−1^ showed the presence of N-H bond and C=C stretching, which may be attributed to the presence of aromatics. Moreover, there are numerous shoulders and small bands in the scope of the region between 1444.55 and 1230.97 cm^−1^ refer to the aromatic rings, C-H bending and C-O stretching vibration absorption peaks [2].

C=O stretching in the alginate extraction was attributed to the intensity of the band at 1391, 1396, and 1397 cm^−1^, while the band at 1632, 1635, and 1628 cm^−1^ remained intense, attributed to interference between the carboxylate and primary amide bands. So, alginate traces in the adsorption capacity of the adsorbents for the removal process can be attributed to the amides. The spectrum of *S. linifolium* polysaccharide contained a short peak at 1239 and 1233 cm^−1^, which was attributed to the presence of sulfated ester groups (S=O), which is a specific component in fucoidan, and at 1032 and 1039 cm^−1^, which corresponded to the S=O stretch of the sulfated polysaccharides or the C-N stretching of aromatic amine group [38]. All these suggested that the polysaccharide of tested seaweeds may be fucoidan and alginate. Table 2 showed that the formation of new peaks, disappearance of some peaks, changes in absorption intensity, or shift in wavenumber of functional groups before and after adsorption, which could be attributed to the interaction of ions in the dye with the active sites of two adsorbents. Furthermore, other studies suggest that there is no noticeable chemical reaction occurring during the adsorption process. As a result, the dye adsorption in this example was most likely electrostatically and physically motivated. The outcome data was compared with the absorption peaks before and after adsorption that is listed in IR spectroscopy, as shown in Table 2.

In that context, the interaction of IV2R with the adsorbent takes place at NH_2_, C=O, COOH, and OH groups, and on the aromatic group existing in the adsorbents, which showed a reduction in those peaks after adsorption.

#### 3.1.2. Raman Spectral Analysis

The vibrational data describe the symmetry of molecules and chemical bonds which could be known from Raman spectroscopy analysis of *A. platensis* (ACDB and LFB), as shown in (Figure 2). Min et al. [7] reported that algae cells mainly comprise five kinds of biomolecules: nucleic acids, carbohydrates, proteins, lipids, and pigments. Each type has its own characteristic signature of Raman spectrum. The absorption peaks at 1644 and 1642 cm^−1^ are due to C=N vibration group and the G band refers to the first-order scattering of the E_2_g phonon of sp^2^ C molecules, while the band at 1513 cm^−1^ is ascribed to the C band. The bands at 73.85–75.01 cm^−1^ correspond to the lattice vibrations in the crystals, which originate from dye ions. The band between 3361.71 cm^−1^ refer to N–H stretching vibrations. The absorption peaks at 3640–4176.30 cm^−1^ are attributed to υ(O-H), while peaks at 1820–1893 cm^−1^ are related to the skeletal vibrations of C=C stretching mode. The absorption peak at 1753–1755 cm^−1^ is representative for C=O groups in carboxyl and carbonyl moieties. In the ACDB, there are disappearances of vibration bands at 3779.79, 3640.87, and 3170.64 cm^−1^; however, the vibration peaks appeared at 4176.30, 3361.14, 3237.91 cm^−1^, 1987.85, 1880.05, 1754.81, 1644.57, and 74.07 cm^−1^, recorded after ACDB adsorption. The formation of new peaks of LFB at 4402.48 and 1155.39 cm^−1^ was caused by –C=N group stretching.

#### 3.1.3. Scanning Electron Microscope

The visualization of the morphology surface of ACDB and LFB before and after adsorption is one of the factors used to evaluate the porosity and average diameter of microalgae particles. The morphology and diameter of the *A. platensis* surface was analyzed by SEM before and after IV2R adsorption (Figure 3). Figure 3A shows the SEM micrograph of ACDB before exposure to dye solution, where cells were agglomerates and had certain dimensions. The surface of ACDB after adsorption showed irregular and uneven surface texture in addition to minor changes of morphology with shrinking and sticking (Figure 3B). Figure 3E shows the SEM image of LFB before adsorption, where particles had certain dimensions, organized and well-shaped. However, the morphology of LFB after adsorption was in the form of a helix, tube, and unfederated distribution with a high porosity (Figure 3F). The change in the cell wall matrix for LFB indicates the increased surface adsorption of dyes. In contrast, the average diameter of ACDB before adsorption of IV2R dye was 194.97 nm, while the mean diameter was 160.69 nm, a little lower than ACDB before adsorption (Figure 3C,D). Figure 3G,H display the diameter distribution of LFB before and after adsorption, which were in the range of 100–260, and 120–240 nm, respectively, with average diameters of 176.09 and 183.87 nm for both LFB before and after adsorption, respectively. The changes in the morphological state and the diameter in LFB may be due to the hydrolysis operation of lipid extraction using concentrated chloroform and methanol. The porosity and pore size of different materials play an essential role for the elimination of dye [30,39].

### 3.2. Influence of Operational Parameters on Adsorption

#### 3.2.1. Adsorbent Dosage

The influence of ACDB and LFB dosage (liquid to solid ratio) on color elimination of the IV2R dye under different biomass dosages was examined (Figure 4). The results showed that the color of dye augmented with the dosage increased from 0.02 to 0.1 g L^−1^ of adsorbent amounts. The percentages of IV2R removal by ACDW adsorbent increased from 56.3 to 84.13%, while using LFB enhanced the removal from 58 to 87.2% using 0.02 g to 0.1 g, respectively. This may be due to a greater availability of surface area or the exchangeable sites at a higher quantity of adsorbent dosage for the complexation of dye ions. In the same way, reduced dye removal is described at very high adsorbent doses owing to incomplete aggregation of adsorbents, in addition to a reduction in the average distance among obtainable adsorption sites [28]. Furthermore, electrostatic interactions between cells could be a major factor in the uptake of biomass-dependent dyes. If the distance between the cells is greater, a huge amount of dye ions is biosorbed; otherwise, a small amount of adsorbent is utilized, which is sufficient for sorption to occur [40]. Even though an increased adsorbent dose has a reducing role on the sorption capacity of a desorbent. Algae have a high surface area and a high binding affinity during biosorption, which increases their adsorption capacity [41]. The algal cell surface contains functional groups such as hydroxyl, carboxylate, amino, and phosphate, which are responsible for the removal of pollutants from wastewater [26].

#### 3.2.2. Initial Dye Concentration

The results of the effect of initial concentrations of dye on the sorption uptake of IV2R are shown in Figure 5. Six initial dye concentrations (10, 20, 30, 40, 50, and 100 mg L^−1^) were used at a constant adsorbent amount of 0.04 g for 2 h at room temperature. As Ismate violet 2R adsorbed relatively well on this medium with a contact time of 120 min, which corresponds to removal rates of 83.3 and 56.4% for ACDB and LFB, respectively, the initial concentration of 10 mg L^−1^ IV2R dye was selected to be used in real wastewater. The number of functional groups available reduced as the initial dye concentration increased. The percentage uptake of dye reduced from 83.3% to 37% and 56.4% to 36.9 1% by augmentation of the dye concentration from 10 to 100 mg L^−1^, for ACDB and LFB, respectively. It has been found that the adsorption rate of IV2R dye is high at the beginning then reduces gradually until the saturation stages were wholly reached after a definite concentration of dye for both the adsorbent biomasses. The number of accessible functional groups reduces as the initial dye concentration increases, as seen by the decrease in percentage sorption as the initial dye ions concentrations increase, due to the fact that the amount of sorbed dye increases in the meantime [8]. This behavior may be due to the fact that at lower concentrations there were sufficient active adsorbent sites for adsorption of presently available dye molecules, but at higher concentrations, due to the inadequate ratio of active sites with dye molecules, adsorption is less [42].

#### 3.2.3. Contact Time mg L^−1^

To assess the optimum contact time requisite to reach the equilibrium state among the liquid phase (effluent) and the solid phase (biomass), this experiment was performed. Figure 6 shows the adsorption removal at different contact adsorption times for the two studied adsorbents. Both the ACDB and LFB adsorbents showed that the percentage removal increased with contact time, and after a certain time, no more dye could be removed from the aqueous solution. The maximum adsorption and equilibrium contact times occurred at 120 min with a percentage removal of 78.9% and 68.26%. This is because there are limited adsorbent sites on the adsorbent and the rate of dye binding with biomass is more predominant during the initial stages; after a definite time, these were exhausted and the adsorption process reached a state of equilibrium and gradually decreased [43].

Vacant surface sites are available at the start of adsorption; nevertheless, once equilibrium is achieved, the remaining vacant sites are difficult to fill, most likely due to repulsive forces between molecules on the adsorbents in the bulk phase [44]. The dye equilibrium can be achieved in less than 15 min using two adsorbents. Still, to be on the safe side, a 120-min shaking time was chosen.

#### 3.2.4. pH Value

The effectiveness with which pollutants are removed from aqueous solutions via adsorption is affected by the pH of the solution, as it affects the features of the adsorbent’s surface charge as well as the form of the contaminants in solution [26]. The effect of initial pH on the percentage of treatment of IV2R using the ACDB and LFB adsorbents were estimated within the pH scope of 2–10 (Figure 7). The results confirmed that the adsorption uptake of LFB increased by the decreased pH, recording the maximum uptake at pH 2 with a percentage removal of 61.7%. However, ACDB dye removal increased with increasing the pH up to 6 with the maximum removal of 70.5%. According to the literature, the complex hetero polysaccharide and lipid elements of algae’s cell wall matrix contain various functional groups such as amino, phosphate, hydroxyl, carboxyl, and other charged groups that are produced by their complex hetero polysaccharide and lipid constituents [45,46,47,48]. So, under acidic conditions, the OH, NH_2_, C=O, and COO groups were protonated; as a result, the ACDB and LFB surface was positively charged. In addition, the ismate violet 2R—three sulphonate groups (D-SO_3_X) were converted to anionic dye ions (D–SO_3_^−^) that make it readily soluble in water, even in an extremely acidic medium [49]. Therefore, in this mechanism, electrostatic attraction happens between the anionic dye’s sulphonate groups and the functional groups on the surface of ACDB and LFB. Additionally, the major adsorption capacity of adsorbent for IV2R adsorption occurs at pH 2–6; at this pH, the amino groups of the adsorbent are protonated, facilitating the adsorption of the negative charged dye, which leads to a decrease in the number of adsorption binding sites for the removal of dye [50]. However, the biomass will have a net positive charge at lower pH levels. It is expected that at acidic pH levels, nitrogen-containing functional groups in the biomass, such as amines or imadazoles, are predicted to be protonated as well. Higher uptakes at lower pH levels might be due to electrostatic interaction between negatively charged dye anions and the positively charged cell surface, as reported by [51,52]. While applying the alga *Stoechospermum marginatum* with acid orange II dye, a reduction in the percentage of dye removal was found with an increase in the pH of the dye solution [53], the adsorption of Reactive Red 120 using the alga *Chara contraria* [54], the removal of Remazol Brilliant Blue R from aqueous solution by the microalgae *Scenedesmus quadricauda* immobilized in alginate gel beads [55], and the removal of Lanaset Red G dye from aqueous solution using the algal species *Chara contraria* [56].

Mahmoud et al. [57] and Carneiro et al. [58] found that under low pH (acidic conditions), hydrogen ions (H+) interact with the amino group (NH_2_) at carbon of the dye structure from NH_3_. This results in the configuration of the ionic form of the dye [C_22_H_15_N_4_O_11_S_3_CuCl]^+^, which has a positive charge on the NH_3_ group. This new ion is less stable. Therefore, as a consequence of the release of two hydrogen atoms from NH3+ in the form of hydrogen gas (H_2_), the positive charge is transferred to carbon.

Nagendrappa [59] found that when the dye is exposed to alkali (high pH), it removes one hydrogen ion (H+) from carbon, which interacts with a hydroxyl group (OH–) from the alkaline media to produce water (H_2_O). As a result, the neutral molecule is transformed into an ionic form with a negative charge at carbon. After that, electrons are transferred from the neighboring sulfonyl group to carbon, forming a structure with a sulfur carbon double bond (S=C) and negatively charged on the sulfonyl group’s oxygen atom. The dye’s change from a neutral component to an ionic form makes it easier to remove from solutions using a variety of ways.

#### 3.2.5. Adsorption Isotherm

The Langmuir isotherm equation is very appropriate to describe the sorption on substance surfaces by homogeneous porosities. Values of the Langmuir constants q_max_ and b obtained from the linear correlation are shown in Figure 8 and Table 3 with the corresponding correlation coefficients “R^2^”. The fitting of investigational results with the Langmuir isotherm was confirmed by high R^2^ values. Additionally, the obtained data demonstrate that the Langmuir isotherm could be applied to provide the adsorption of IV2R dye by ACDB and LFB with the correlation coefficient “R^2^” of 0.979 and 0.994 for both ACDB and LFB adsorbents, respectively. This means that the Langmuir model could be applicable to the equilibrium data for the sorption of IV2R. Furthermore, the maximal uptake capacity (q_max_) is important to identify the adsorbent highest dye uptake. The maximal uptake capacity (q_max_) was 14.70 and 9.90 mg g^−1^ for ACDB and LFB adsorbents, respectively. The Langmuir constant (b), which reflects the heat of the sorption, was examined to be 0.12 and 0.023 for removal of IV2R for both ACDB and LFB adsorbents. The essential characteristics of the Langmuir model can be calculated via the constant of dimensionless so-called equilibrium parameter, (R_L_), defined through the following equation [60]:R_L_ =1/(1 + Ka × C_i_)(15)

R_L_ can serve to demonstrate the affinity among the sorbate (dye ions) and sorbent (ACDB and LFB) by separation factor or/and dimensionless equilibrium factor and Ka is the constant of Langmuir. The results of separation factor R_L_ gives important information around the nature of adsorption. From this study, the average R_L_ was obtained to be 0.877 and 0.994 for concentration of 10–100 mg L^−1^ of IV2R dyes for ACDB and LFB, respectively; as displayed in Table 3, the value data of R_L_ indicated the type of Langmuir equation to be favorable (0 < R_L_ < 1), irreversible (R_L_ = 0), unfavorable (R_L_ > 1), and linear (R_L_ = 1) [35]. They are in the scope from 0–1, which describes the favorable adsorption process.

Freundlich isotherm gave a better fit for IV2R with a higher correlation coefficient of 0.988 for LFB adsorbent, while this model gave a lower fit by ACDB adsorbent with correlation coefficient 0.856 as shown in Figure 9 and Table 3. On the other hand, the Freundlich constant “K_f_” displays the sorption capacity on heterogeneous sites with non-uniform spreading of the energy level and can work as indicators for the maximum adsorption capacity of the dye cation from the adsorbent. The value of the “n” parameter of the Freundlich equation can act as an indicator on the favorability of adsorption and shows the intensity among dyes and biomass. The “K_f_” values were 5.76 and 6.3 for both ACDB and LFB adsorbents, respectively. Moreover, a strong bond is existent among two adsorbents, as shown via the data of 1/n, “called the heterogeneity factor”, describing the deviation from the linearity of adsorption as follows: when 1/n is below 1, chemical adsorption happens, and this demonstrates an ordinary Langmuir isotherm; however, when 1/n is further than 1, cooperative adsorption takes place, and the sorption process is more favorable physically and comprises strong interactions between the particles of the dye solution; when 1/n is equivalent to 1, the adsorption is linear, and the dye particle concentration does not influence the division between the two stages [34]. In this study, the values of factor “1/n” are less than 1 for ACDB and LFB adsorbents; the values show a physical adsorption method on an external surface with this equation to be favorable.

Tempkin isotherm shows the influence of some indirect interactions between adsorbate molecules and suggests a linear reduction in the heat of the sorption process of the particles in each layer, owing to these interactions [61]. The value of the regression coefficient (R2+) was found to be 0.67 and 0.881 for both ACDB and LFB adsorbents, respectively (Figure 10), which indicate that this model is not favorable for the removal of IV2R dye. The Tempkin constants A and B were presented in Table 3.

The Halsey isotherm is appropriate for multilayer sorption process, and the fitting of the Halsey model could be useful to heterologous solids [62]. The plots of Ln q_e_ against the Ln C_e_ Halsey adsorption isotherms are shown in Figure 11. The parameters obtained for the Halsey isotherm were fitted with ACDB and LFB adsorbents, with high regression correlation coefficients ranging between 0.856 and 0.989. The Halsey isotherm parameters of K and n are presented in Table 3.

The kinetics study of adsorbate uptake is an important parameter for selecting the best operating conditions to know the dynamics of the sorption reaction and for design purposes in expressions of the order of the rate constant. The kinetics values achieved from the sorption process of IV2R dye onto ACDB and LFB was examined by using three communal kinetic models, which are the pseudo-first- and second-order kinetic models in addition intraparticle diffusion model. Furthermore, the most appropriate model was chosen depending on the regression correlation coefficient (R^2^) to determine how well the predicted data from a forecast model matches with the investigational data. The pseudo-first order was applied to explain the sorption rate depending on the capacity of adsorption. This model posits that the ratio of occupation of adsorption sites in the surface is proportional to the numeral of unoccupied sites. Table 4 provided the result data of K_1_, investigational and calculated data of q_e_, in addition to the correlation coefficients value for the pseudo-first kinetic plots. The theoretical values of q_e_ are not in line with the experimental data acquired. This suggests a poor fit amongst the kinetics data and the pseudo-first order equation. Additionally, it was presented that the pseudo-first order were ruled out as a result of their correlation coefficients for the current experimental results being small (R^2^ = 0.113 and 0.716) for ACDB and LFB adsorbents, respectively. This suggests that this adsorption system is not an ideal first-order reaction. It was observed from Figure 12 and Table 4 that the pseudo-second order kinetic fit well with higher correlation coefficients (R^2^ = 0.980 and 0.987 for ACDB and LFB adsorbents, respectively). Moreover, it was clear that the result value of k_2_ parameter was greater than the corresponding k1 parameter value. This is for the reason that the pseudo-second-order presumes that the adsorption rate is relative to the square of a number of unoccupied sites, as reported by Gupta et al. [63].

The figures of q_t_ inverse t^0.5^ explain the intraparticle diffusion model, which may represent a numerous-linearity correlation, which shows that two or more stages happen through the adsorption method (Figure 13). Moreover, the rate constant K_dif_ directly estimated from the slope and the intercept is C, as recorded in Table 4. The values’ C factors offer information around the thickness of the border layer, as the resistance to the exterior mass transfer rises as the intercept increases. Furthermore, the low linearity of the plots demonstrated intra-particle diffusion in the uptake of IV2R dyes by ACDB and LFB with a low correlation coefficient R^2^.

#### 3.2.6. Applicability on Actual Wastewater

To evaluate the validity of the uses of ACDB and LFB as adsorbents, real wastewater samples were collected to examine for removal of IV2R by two adsorbents studies under optimization conditions. The application of Spirulina to eliminate IV2R from wastewater was carried out under optimization conditions. Table 5 shows that a near 75.7 and 61% removal of IV2R dye from real wastewater was detected for ACDB and LFB, respectively, while the percentage elimination of IV2R dye from aqueous solution was 83 and 56.4%. The results confirmed that the percentage removal of IV2R through two adsorbents from aqueous solution was not influenced by replacing the distilled water by real wastewater in LFB, while the highest removal for ACDB was significantly altered by changing of the kind of water, of which deionized water exhibited the lowest influence on the adsorption method. Conversely, the real wastewater effluents can comprise very high concentrations of interfering ions from numerous pollutants that will have a significant influence on the elimination efficiency of IV2R dye.

### 3.3. Rotifer Bioassay

Marine Rotifera, *B. plicatilis*, is a zooplankton that is extensively utilized as live feed for larvae in aquatic hatcheries [22,64]. Interestingly, rotifer (*Brachionus plicatilis*) is one of the most extensively aquatic animals utilized in environmental toxicity studies due to it being easy to keep in the laboratory and that it can be preserved at high population densities in minor volumes, has a short life cycle, and that it is a good indicator of heavy metal pollution [65]. In the current study, Figure 14A–D shows the rotifer population, rotifer mortality, female carry eggs population, and the mortality of female carry eggs, respectively.

At all levels, the ACDB and LFB loaded by IV2R resulted in lower rotifer *B. plicatilis* population (Figure 14A) and the number of female carry eggs (Figure 14B), when compared to the traditional ACDB and LFB (controls) at the same levels. On the contrary, ACDB and LFB loaded by IV2R resulted in higher rotifer total mortality (Figure 14C) and female carry eggs mortality (Figure 14D), when compared to the traditional ACDB and LFB at the same levels. In the case of ACDB and LFB loaded by IV2R, the level 0.02 LFB has achieved the highest rotifer population, followed by the level of 0.02 ACDB. The levels 0.02 LFB and 0.2 ACDB achieved the highest number of female carry eggs. The higher mortality of total rotifer number and/or female carry eggs mortality were observed in higher levels of ACDB and LFB loaded by IV2R, compared to those of lower levels of ACDB and LFB loaded by IV2R. These results concluded that marine rotifer *B. plicatilis* was sensitive to dye IV2R attributed to reduced rotifer population and female carry eggs and increased rotifer mortality, in general. Due to the attendance of metal ions of dyes from organic dyes in the marine environment influencing the marine life, which can result in the death of zooplankton or slow growth organisms that, based on zooplankton for food and also dyes, can decrease zooplankton nutrition supplies of larval fish [23].

## 4. Conclusions

The current work investigated the efficiency of ACDB and LFB in bioremediation of dye (IV2R) from industrial textile effluents. The potential of the ACDB and LFB loaded by IV2R as a feed for rotifer *B. plicatilis* throughout the bioassay test was also investigated. The adsorption of IV2R dye into *Arthrospira platensis* NIOF17/003 was prepared and the given results showed that the adsorption process increased with increasing ACDB and LFB dose, contact time (120 min), initial IV2R concentration (10 mg L^−1^), and acidity pH (2 and 6, respectively), giving it a relatively large affinity with respect to ACDB and LFB. Physical and chemical characteristics of ACDB and LFB were obtained by FTIR, SEM, and Raman spectroscopy. Moreover, Freundlich, Langmuir, Tempkin, and Halsey adsorption models were conducted to describe the equilibrium isotherms while the isotherm constants were determined. By comparing the correlation coefficients obtained for the studied isotherm models, all isotherm models fit the experimental data reasonably well, except Tempkin models for LFB. In addition, Langmuir was fit for ACDB at optimum condition with a regression co-efficient higher than 0.9. However, the value of R_L_ in the case of Langmuir isotherms lies between 0 and 1, indicating convenient adsorption. In addition, the three models (pseudo-first order, pseudo-second order, and intra-particle diffusion) were conducted to find the suitable kinetic model for adsorption techniques. It has been found that the sorption kinetics of IV2R has a poor fit with the pseudo-first-order kinetic equation, while they have preferably obeyed the second-order kinetic models which provide a higher correlation coefficient. Furthermore, the intra-particle diffusion was also examined, and it was revealed that the adsorption process is regulated by the film diffusion with a regression co-efficient higher than 0.82 and 0.89 for ACDB and LFB, respectively. For the removal of industrial textile effluents, the ACDB and LFB adsorbents offered a good ability to eliminate 75.7% and 61.11%, respectively, of a dye solution. On the other hand, as a by-product, the ACDB and LFB loaded by IV2R up to a level of 0.02 g could be used as feed for marine rotifer *B. plicatilis*. Furthermore, the findings indicated that adsorbent may be successfully utilized to extract IV2R dye from aqueous solutions and wastewater at a cheap cost and with minimal environmental impact.

## Figures and Tables

**Figure 1 materials-14-04446-f001:**
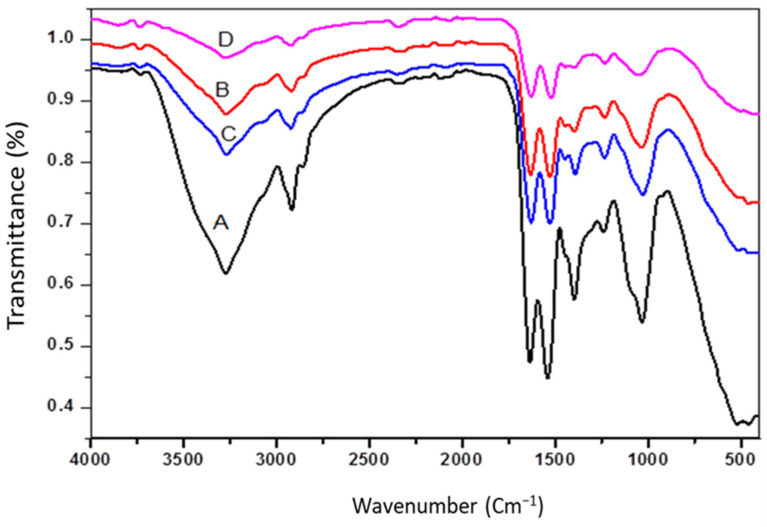
FTIR spectra: (A) ACDB before adsorption; (B): ACDB after adsorption; (C) LFB before adsorption; (D) LFB after adsorption.

**Figure 2 materials-14-04446-f002:**
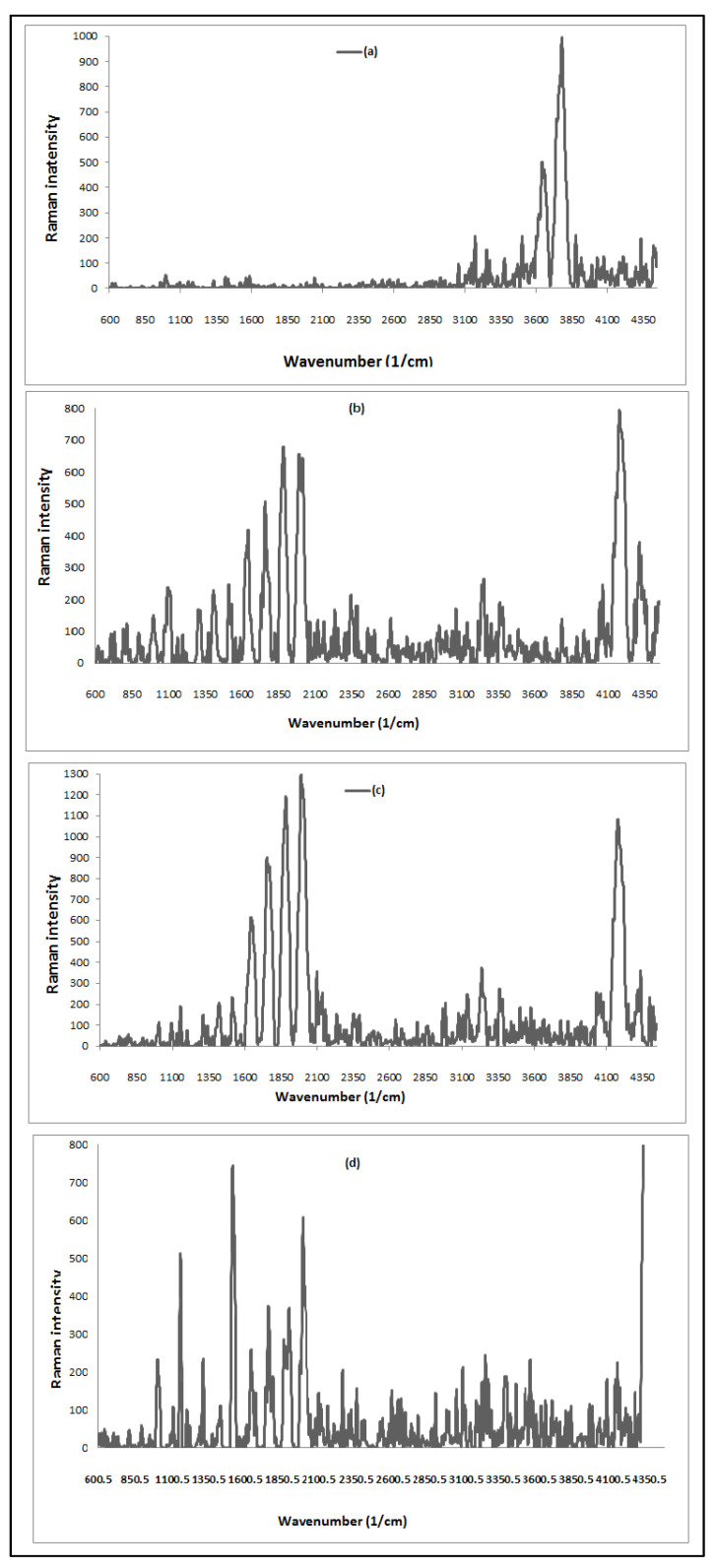
Raman spectrum for spectra for: ACDB before adsorption (**a**); ACDB after adsorption (**b**); LFB before adsorption (**c**); LFB after adsorption of IV2R dye (**d**).

**Figure 3 materials-14-04446-f003:**
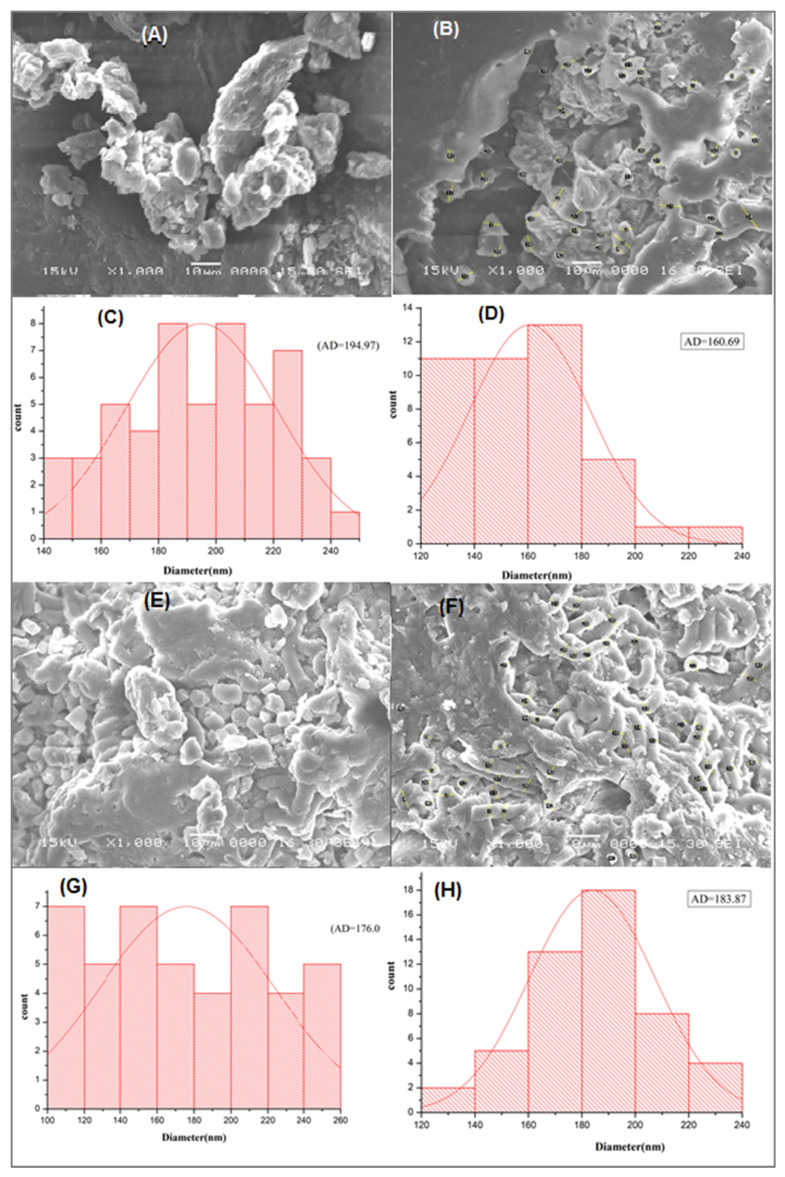
SEM pictures of ACDB before adsorption (**A**), after adsorption (**B**), and LFB before adsorption (**E**); LFB after adsorption (**F**), and the corresponding diameter distributions of (**C**,**D**,**G**, and **H**, respectively).

**Figure 4 materials-14-04446-f004:**
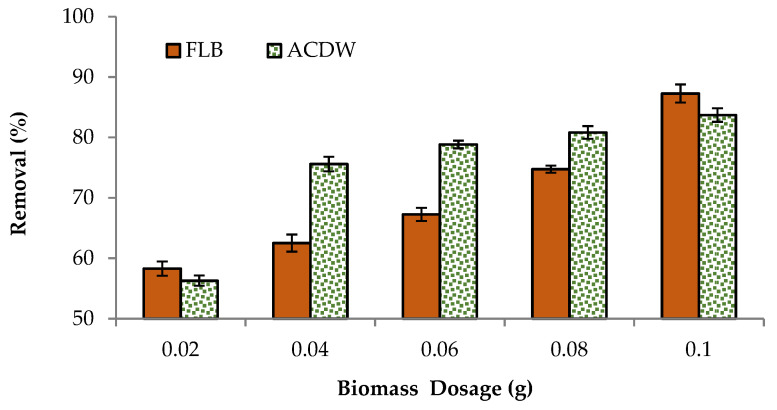
Influence of adsorbent dosage on the elimination efficiency of IV2R dye.

**Figure 5 materials-14-04446-f005:**
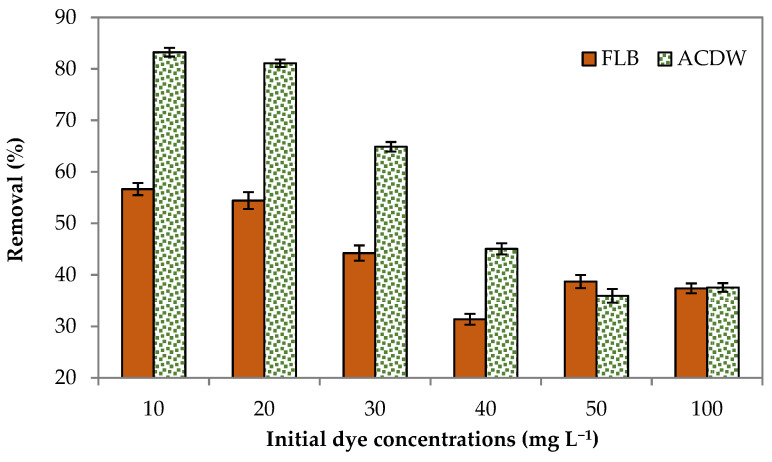
Influence of initial dye concentrations on elimination efficiency of IV2R dye.

**Figure 6 materials-14-04446-f006:**
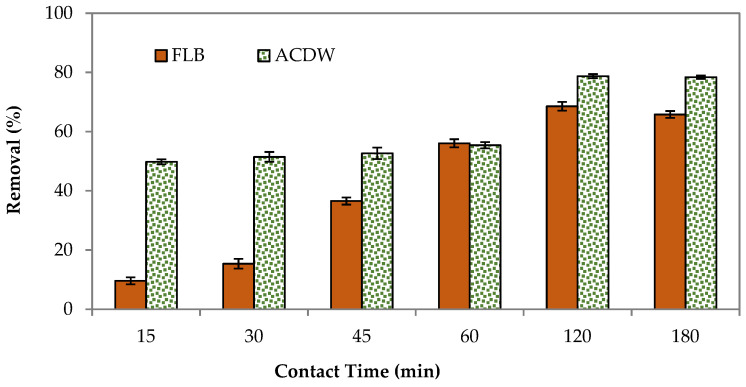
Influence of contact time on elimination efficiency of IV2R dye.

**Figure 7 materials-14-04446-f007:**
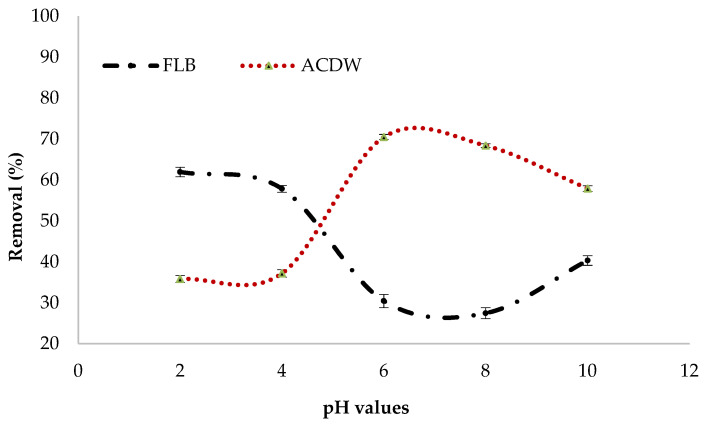
Influence of pH values on elimination efficiency of IV2R dye.

**Figure 8 materials-14-04446-f008:**
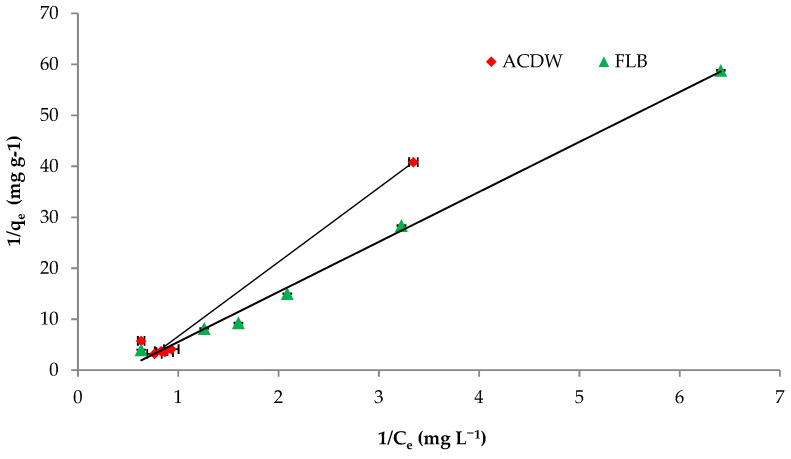
Langmuir isotherm plot for removal efficiency of IV2R dye.

**Figure 9 materials-14-04446-f009:**
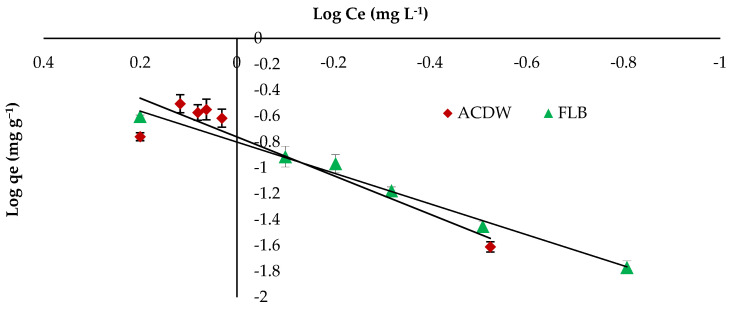
Freundlich isotherm plot for removal efficiency of IV2R dye.

**Figure 10 materials-14-04446-f010:**
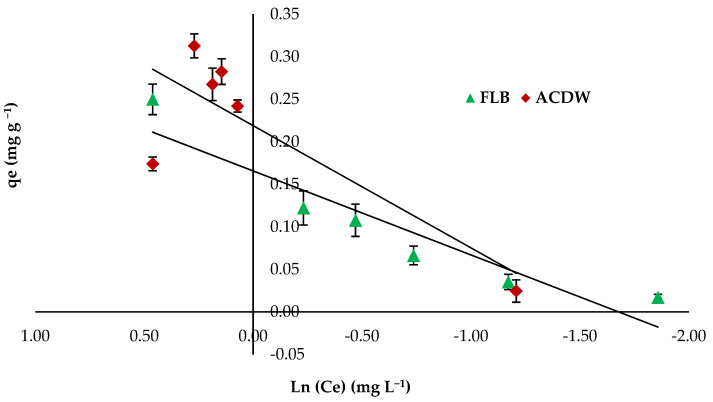
Tempkin isotherm plot for removal efficiency of IV2R dye.

**Figure 11 materials-14-04446-f011:**
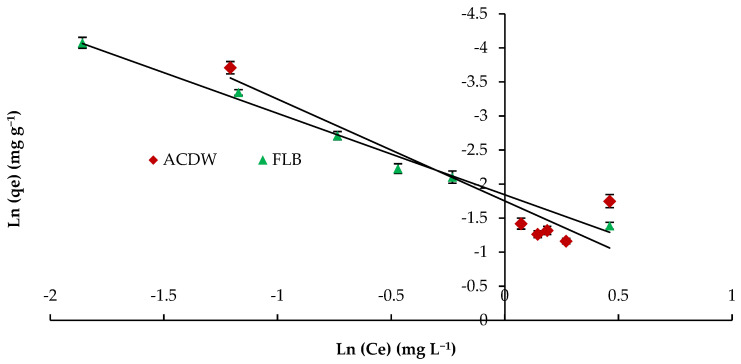
Halsey model plot for removal efficiency of IV2R dye.

**Figure 12 materials-14-04446-f012:**
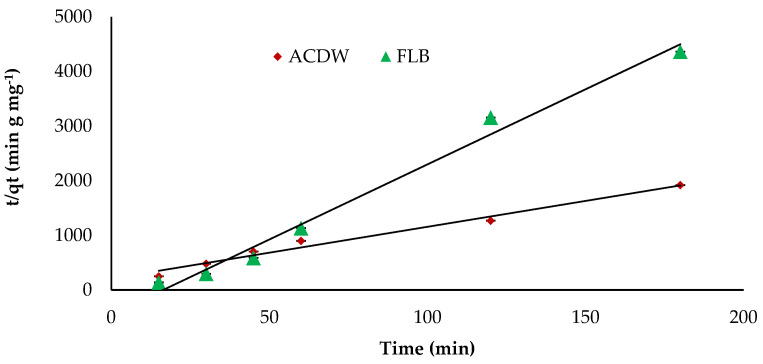
Pseudo-second kinetics of IV2R dyes onto ACDB and LFB.

**Figure 13 materials-14-04446-f013:**
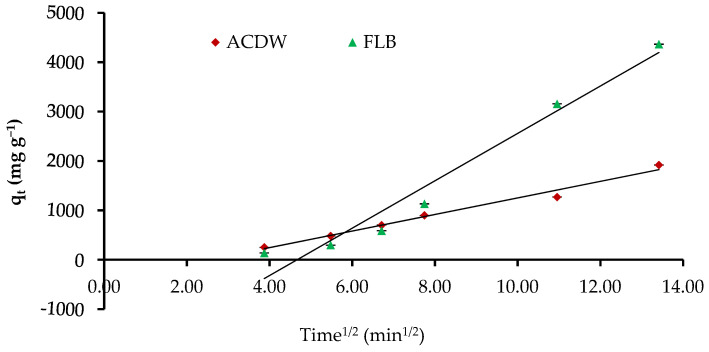
Intraparticle diffusion kinetics of IV2R dyes onto ACDB and LFB.

**Figure 14 materials-14-04446-f014:**
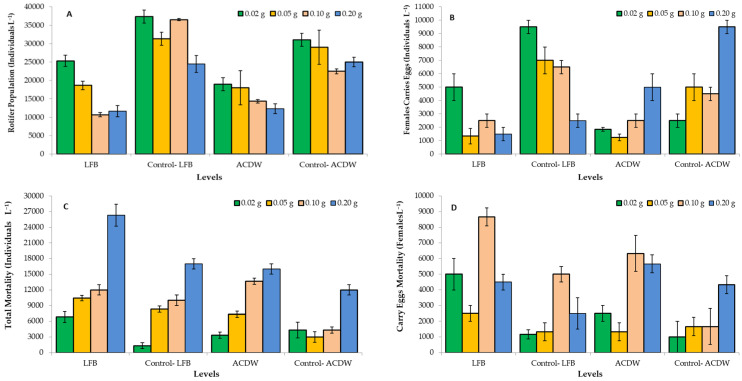
Effect of different levels of *Arthrospira platensis* NIOF17/003 complete dry weight (ACDB) and free lipid (LFB) which was loaded by dye (Ismate violet 2R, IV2R) removed from industrial textile effluents, on rotifer *Brachionus plicatilis* population (**A**), female carry eggs population (**B**), and total rotifer mortality (**C**), and female carry eggs mortality (**D**).

**Table 1 materials-14-04446-t001:** characteristics of physical and chemical for ISMATE violet 2R [30].

Characteristics	Value
Dye name (a common name)	Ismate violet 2R
Wavelength (λ max)	550 nm
Mol. wt.	700
Molecular formula	C_22_H_14_N_4_O_11_S_3_CuCl
Color Index name	IV2R
Molecular structure	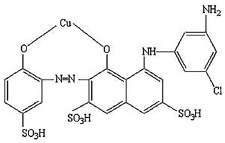

**Table 2 materials-14-04446-t002:** FT-IR spectrum of the tested ACDB and LFB, before and after adsorption.

ACDB		LFB	
Before Adsorption (cm^−1^)	After Adsorption (cm^−1^)	Before Adsorption (cm^−1^)	After Adsorption (cm^−1^)
457.77	1034.97	448.73	1047.13
1032.15	1233.58	1029.56	1230.93
1239.13	1396.38	1233.22	1397.83
1396.98	1443.85	1391.72	1444.55
1540.10	1526.94	1446.89	1521.77
1635.36	1632.08	1527.21	1628.89
2917.13	2329.33	1628.75	2067.80
3273.47	2929.06	2921.64	2349.79
3730.69	3270.67	3269.13	2922
-	3734.47	3732.82	3271.80
-	3847.56	-	3735.70
		-	3856.80

**Table 3 materials-14-04446-t003:** Values of isotherm constants for IV2R dye removal by ACDB and LFB.

Isotherm Model	Isotherm Parameter	Values	
		ACDB	LFB
Freundlich	n	0.668	0.837
	K_F_	5.766	6.3
	R^2^	0.856	0.988
Langmuir	q_max_ (mg g^−1^)	14.70	9.90
	b	0.129	0.023
	R^2^	0.979	0.994
	R_L_	0.856	0.988
Tempkin	A_T_	11.42	5.36
	B_T_	0.143	0.098
	b_T_	0.195	0.284
	R^2^	0.67	0.881
Halsey	1/n	1.49	1.19
	K	1.75	1.84
	R^2^	0.856	0.989

**Table 4 materials-14-04446-t004:** Comparison of the kinetic model for IV2R dyes from ACDB and LFB.

Model	1st-Order Kinetic Model			2nd-Order Kinetic Model			Intraparticle Diffusion Model		
Parameters	q_e_ (calc.) (mg g^−1^)	k_1_ (1 min^−1^)	R^2^	q_e_ (calc.) (mg g^−1^)	k_2_ (mg g^−1^ min^−1^)	R^2^	K_dif_ (mg g in^−0.5^)	C	R^2^
ACDB	13.18	0.012	0.11	0.105	0.426	0.980	0.0078	0.123	0.825
LFB	1.018	9.21 × 10^−4^	0.716	0.036	1.688	0.987	0.0043	0.039	0.891

**Table 5 materials-14-04446-t005:** Percentage of removal for IV2R dyes by using ACDB and LFB in a different water sample.

Types of Water	IV2R Removal (%)	
	ACDB	LFB
Industrial wastewater	75.79	61.11
Distilled water	83.33	56.41

## Data Availability

All relevant data are within the paper, and are available from the corresponding author.

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
