# Peer review of "Potential Applications of Arthrospira platensis Lipid-Free Biomass in Bioremediation of Organic Dye from Industrial Textile Effluents and Its Influence on Marine Rotifer (Brachionus plicatilis)"

_materials, 2021, doi:10.3390/ma14164446_

Round 1
Reviewer 1 Report
It is suggested that all comments to be considered in the revised manuscript, as followed.
1. page 3 :
- What is the reason that Ismate violate 2R was chosen in this study, as there are so many kinds of organic dyes ? In addition what kind of this dye as a classification of organic dyes ?
2. Table 2 :
- The chemical structure of dye(Ismate violate 2R) is not clear, therefore it should be improved by its quality.
- "CI Name" as IV2R should be changed to its full Colour Index name. "Ismate violate 2R" can be assumed as a common name, but the official scientific name, such as a Colour Index name, should be used in the research paper, and "violate" to be "violet".
3. page 7 & 11~12 :
- It is recommended to provide the presumed interaction mechanism between a dye and adsorbent which would be best added as a figure with additional discussions. Since the dye contains both acidic groups(-SO3H) and basic groups(-NH2), thus the pH value can be very critical to form the interactions. For the detailed discussions, the chemical structures of adsorbent(ACBD & LFB) also clearly provide. The discussions regarding to the FT-IR Spectra in page 7 should be improved by the presumed mechanism.
4. Figure 1 : It is not apparent the differences in peaks for two adsorbents between before and after adsorption.
Author Response
SUMMARY OF AUTHOR(S) RESPONSE TO REVIEWER’S COMMENTS
Manuscript Title: Potential Applications of Arthrospira platensis lipid-free biomass in Bioremediation of Organic Dye from Industrial Textile Effluents and its Influence on Marine Rotifer (Brachionus plicatilis).
Authors: Ahmed E. Alprol, Ahmed M.M. Heneash, Mohamed Ashour, Khamael M Abualnaja, Dalal Alhashmialameer, M.A. Abu-Saied, Abdallah Tageldein Mansour, Zaki Z. Sharawy, Abd El-Fatah Abomohra
|
Reviewer 1# Round 1 |
Author(s) response |
|
Comments: It is suggested that all comments to be considered in the revised manuscript, as followed.
|
|
|
Page 3 : - What is the reason that Ismate violate 2R was chosen in this study, as there are so many kinds of organic dyes? In addition what kind of this dye as a classification of organic dyes? |
- Ismate violet 2R was selected as a model compound in this research due to its wide application range, which includes coloring the silk, cotton, rayon, leather, paper ,wood and coating for paper stock and medical purposes, in addition to its potential dangerous effects. Additionally, the Ismate violet 2R classified as sulfur dye according to their application (Page: 3, Line: 115-125). |
|
Table 1 : - The chemical structure of dye (Ismate violate 2R) is not clear, therefore it should be improved by its quality. |
- In table 1; the chemical structure of dye (Ismate violate 2R) was improved (Page: 4, Line: 170). |
|
- "CI Name" as IV2R should be changed to its full Colour Index name. "Ismate violate 2R" can be assumed as a common name, but the official scientific name, such as a Colour Index name, should be used in the research paper, and "violate" to be "violet". |
- Corrected in manuscript (Table 1) (Page:4, Line: 170). |
|
page 7 & 11~12 : - It is recommended to provide the presumed interaction mechanism between a dye and adsorbent which would be best added as a figure with additional discussions. Since the dye contains both acidic groups (-SO3H) and basic groups (-NH2), thus the pH value can be very critical to form the interactions. For the detailed discussions, the chemical structures of adsorbent (ACBD & LFB) also clearly provide.
|
- Further results, discussion and references were added in manuscript, in addition to the supposed interaction mechanism between a dye and adsorbent was added in section 2.2. Influence of operational parameters on adsorption, as in the lines shaded yellow (Page: 12, Lines: 393-401; Page: 13, Lines: 435-443; Page: 14, Lines: 445-447; 456-469). |
|
- The discussions regarding to the FT-IR Spectra in page 7 should be improved by the presumed mechanism. - Figure 1: It is not apparent the differences in peaks for two adsorbents between before and after adsorption. |
- The discussions improved by the presumed mechanism as in the lines shaded yellow (Page: 7, Lines: 307-309& Page: 8, Lines 313-325). |
|
- Table (2) was added to show the formation of new peaks, disappearance of some peaks, changes in absorption intensity, or shift in wavenumber of functional groups before and after adsorption (Page: 8, Lines: 335 - 336). |
|
We would like to extend our sincere thanks and appreciation to the reviewers and editorial board. In fact, their comments and guidance added a lot to the research and increased its scientific content. Therefore, the words cannot express their gratitude for their time and effort they put in evaluating this research.
Reviewer 2 Report
The authors present a complete experimental study on the efficiency of Arthrospira-Complete-Dry-Biomass and lipid-free biomass in the bioremediation of dye from textile effluents.
The authors make use of SEM, FTIR, and Raman techniques to characterize the surface of the adsorbents with the aim to understand the adsorption mechanism by also employing proper theoretical modeling and corresponding statistical analysis.
The results are sound allowing to understand the influence of adsorbent dosage and of initial dye concentration on the elimination efficiency of the considered dye. Furthermore, the authors showed that the obtained by-product could be used as feed for marine rotifer B. plicatilis. Conclusions are supported by the achieved information so that the work would deserve to be published.
However, the paper needs minor revisions before it can be considered for publication: for example, it contains several English errors or misprints and stylist issues to be fixed:
- English errors or misprints occur throughout the paper and careful reading is needed: see for instance the x-axis label of figures 1 and 5; the sentence between lines 302-303 needs to be corrected; line 371 “…adsorption process attained an equilibrium state and gradually decreases” you should use the same verb tense, etc., etc.
- Figure captions and main text should be more spaced and references contain too many full stops (to be fixed at least during proofreading)
- Figures plot, axis labels, and figure captions are not well arranged. For instance, note figure 13 (but almost all figures are not correctly displayed): the unit measure of the y-axis label is not well reported, the plot and the x-axis tick labels badly superimpose.
- Raman spectra shown in figure 2 and the histograms in figure 3 should be reported on the same scale for a better comparison.
- Which are the experimental errors on the achieved results and so on the corresponding values reported in figures 8-13?
- Too many self-citations are reported. For instant Ashour, M. is present on 24/56 references.
Author Response
SUMMARY OF AUTHOR(S) RESPONSE TO REVIEWER’S COMMENTS
Manuscript Title: Potential Applications of Arthrospira platensis lipid-free biomass in Bioremediation of Organic Dye from Industrial Textile Effluents and its Influence on Marine Rotifer (Brachionus plicatilis).
Authors: Ahmed E. Alprol, Ahmed M.M. Heneash, Mohamed Ashour, Khamael M Abualnaja, Dalal Alhashmialameer, M.A. Abu-Saied, Abdallah Tageldein Mansour, Zaki Z. Sharawy, Abd El-Fatah Abomohra
|
Reviewer 2# Round 1 |
Author(s) response |
|
Comments and Suggestions for Authors |
|
|
The authors present a complete experimental study on the efficiency of Arthrospira-Complete-Dry-Biomass and lipid-free biomass in the bioremediation of dye from textile effluents.The authors make use of SEM, FTIR, and Raman techniques to characterize the surface of the adsorbents with the aim to understand the adsorption mechanism by also employing proper theoretical modeling and corresponding statistical analysis. The results are sound allowing to understand the influence of adsorbent dosage and of initial dye concentration on the elimination efficiency of the considered dye. Furthermore, the authors showed that the obtained by-product could be used as feed for marine rotifer B. plicatilis. Conclusions are supported by the achieved information so that the work would deserve to be published. However, the paper needs minor revisions before it can be considered for publication. |
- The authors would like to thank Reviewer # 2 for his kind and his interesting and valuable comments. All Reviewer # 2 comments have been considered carefully by the authors. These comments significantly improve the manuscript. |
|
English errors or misprints occur throughout the paper and careful reading is needed: for instance the x-axis label of figures 1 and 5. |
- The x-axis label of figures 1 and 5 was corrected (Pages: 8&13, Lines: 330&423). As well as, all manuscript was revised well. |
|
The sentence between lines 302-303 needs to be corrected; line 371 “…adsorption process attained an equilibrium state and gradually decreases” you should use the same verb tense, etc. |
- This sentence was corrected (Page: 13, Line: 433).
|
|
Figure captions and main text should be more spaced and references contain too many full stops (to be fixed at least during proofreading). |
- Thank you for your observation. It will be fixed during the proofreading. |
|
Figures plot, axis labels, and figure captions are not well arranged. For instance, note figure 13 (but almost all figures are not correctly displayed): the unit measure of the y-axis label is not well reported, the plot and the x-axis tick labels badly superimpose. |
- All figures have been redrawn to clearer and figure captions are arranged, in addition the unit measure of the x & y-axis label was added. |
|
Raman spectra shown in Figure 2 and the histograms in Figure 3 should be reported on the same scale for a better comparison |
- Raman spectra shown in Figure 2 and the histograms in Figure 3 should be reported on the same scale for a better comparison |
|
Which are the experimental errors on the achieved results and so on the corresponding values reported in figures 8-13? |
- The experimental errors (SD) on the achieved results were added in figures 8-13. |
|
Too many self-citations are reported. For instant Ashour, M. is present on 24/56 references. |
- We are so sorry about that, it has been reduced. |
We would like to extend our sincere thanks and appreciation to the reviewers and editorial board. In fact, their comments and guidance added a lot to the research and increased its scientific content. Therefore, the words cannot express their gratitude for their time and effort they put in evaluating this research.
Reviewer 3 Report
The article needs to enrich the review with information on the importance of post-industrial pollution in inland waters and marine areas. In my opinion, it is worth pointing out the aspect of accumulation of harmful compounds in the environment.
- In the literature review, I am missing information regarding the allowable concentrations of chemicals in industrial wastewater.
- Section 1.2 lacks information on what the permitted dye content is or what amounts of dye are actually present in the waste water. In brief, on what principle were the concentrations selected
- Has a study been conducted demonstrating the presence or absence of accumulation of dye/derived products from dye in B. plicatilis?
Conclusion:
In my opinion, the proposals should indicate a more universal pacing range. For example. using the xyz model allows to determine the AAA relationship.
Author Response
SUMMARY OF AUTHOR(S) RESPONSE TO REVIEWER’S COMMENTS
Manuscript Title: Potential Applications of Arthrospira platensis lipid-free biomass in Bioremediation of Organic Dye from Industrial Textile Effluents and its Influence on Marine Rotifer (Brachionus plicatilis).
Authors: Ahmed E. Alprol, Ahmed M.M. Heneash, Mohamed Ashour, Khamael M Abualnaja, Dalal Alhashmialameer, M.A. Abu-Saied, Abdallah Tageldein Mansour, Zaki Z. Sharawy, Abd El-Fatah Abomohra
|
Reviewer 3# Round 1 |
Author(s) response |
|
Comments: |
|
|
The article needs to enrich the review with information on the importance of post-industrial pollution in inland waters and marine areas. In my opinion, it is worth pointing out the aspect of accumulation of harmful compounds in the environment. In the literature review, I am missing information regarding the allowable concentrations of chemicals in industrial wastewater. |
-The introductory part has been supported with information regarding the allowable concentrations of chemicals in industrial wastewater in addition to literature review (Page: 2, Lines: 57-72).
|
|
Section 1.2 lacks information on what the permitted dye content is or what amounts of dye are actually present in the waste water. In brief, on what principle were the concentrations selected. |
- The concentration of dye are actually present in the waste water was selected according to previous studies in the field of dye removal (Ref. 45), in order to test the possibility of removing it at different concentrations in order to choose the best concentration (optimization condition) as it can be applied to assessment the effect of other factors as well as for its application on real wastewater. - While, after the process of choosing the best conditions, the concentration 10 mg L-1 was chosen because it achieved the largest percentage of dye removal (Section: Page: 4, Line: 178-180). |
|
Has a study been conducted demonstrating the presence or absence of accumulation of dye/derived products from dye in B. plicatilis? |
- Actually, so far, not yet, while we will do it in the future study. |
|
Conclusion: In my opinion, the proposals should indicate a more universal pacing range. For example, using the xyz model allows to determine the AAA relationship. |
- The conclusion section has been improved, revised, rewritten, and supported with the principal results, and the best fit kinetic model for both isotherm and kinetic models was added in conclusion (Page: 20 & 21, Lines: 640,642-655; 659-660). |
We would like to extend our sincere thanks and appreciation to the reviewers and editorial board. In fact, their comments and guidance added a lot to the research and increased its scientific content. Therefore, the words cannot express their gratitude for their time and effort they put in evaluating this research.
Round 2
Reviewer 1 Report
Most of comments were considered properly to the revised manuscript, nevertheless it is not clear the formation of hydrogen bonds between -OH groups and an oxygen atom of -SO3H which should be affected by pH conditions. Therefore more detailed discussions required in 14 page including pH effect as shown equation (17) & (18).
Author Response
The authors extend their sincere thanks and appreciation to the anonymous reviewer. In fact, his comments and guidance added valuable value to the manuscript and increased its scientific content. Therefore, words cannot express our gratitude for the time and effort he put into evaluating this manuscript.
All suggested comments of Reviewer #1 in Round 2 have been conducted (in Green shadow) as requested on Page 14, Lines: 445-482.
Best Regards
Authors
